# Undescribed Cyclohexene and Benzofuran Alkenyl Derivatives from *Choerospondias axillaris*, a Potential Hypoglycemic Fruit

**DOI:** 10.3390/foods13101495

**Published:** 2024-05-11

**Authors:** Ermias Tamiru Weldetsadik, Na Li, Jingjuan Li, Jiahuan Shang, Hongtao Zhu, Yingjun Zhang

**Affiliations:** 1State Key Laboratory of Phytochemistry and Plant Resources of West China, Kunming Institute of Botany, Chinese Academy of Sciences, Kunming 650201, China; ermiastamiruweldetsa@mail.kib.ac.cn (E.T.W.); lina1@mail.kib.ac.cn (N.L.); lijingjuan@mail.kib.ac.cn (J.L.); shangjiahuan@mail.kib.ac.cn (J.S.); zhuhongtao@mail.kib.ac.cn (H.Z.); 2University of Chinese Academy of Sciences, Beijing 100049, China; 3Yunnan Key Laboratory of Natural Medicinal Chemistry, Kunming Institute of Botany, Chinese Academy of Sciences, Kunming 650201, China

**Keywords:** *Choerospondias axillaris*, fruits, alkenyl and benzofuran derivatives, *α*-glucosidase inhibitory activity, molecular docking

## Abstract

The fruit of *Choerospondias axillaris* (Anacardiaceae), known as south wild jujube in China, has been consumed widely in several regions of the world to produce fruit pastille and leathers, juice, jam, and candy. A comprehensive chemical study on the fresh fruits led to the isolation and identification of 18 compounds, including 7 new (**1**–**7**) and 11 known (**8**–**18**) comprised of 5 alkenyl (cyclohexenols and cyclohexenones) derivatives (**1**–**5**), 3 benzofuran derivatives (**6**–**8**), 6 flavonoids (**9**–**14**) and 4 lignans (**15**–**18**). Their structures were elucidated by extensive spectroscopic analysis. The known lignans **15**–**18** were isolated from the genus *Choerospondias* for the first time. Most of the isolates exhibited significant inhibitory activity on *α*-glucosidase with IC_50_ values from 2.26 ± 0.06 to 43.9 ± 0.96 μM. Molecular docking experiments strongly supported the potent *α*-glucosidase inhibitory activity. The results indicated that *C. axillaris* fruits could be an excellent source of functional foods that acquire potential hypoglycemic bioactive components.

## 1. Introduction

*Choerospondias axillaris* (Roxb.) B.L. Burtt. & A.W. Hill (Anacardiaceae), a tall deciduous tree bearing edible fruits, is distributed in slopes, hills or valleys at altitudes ranging from 300 to 2000 m in India, China, Japan, Bhutan, Laos, Vietnam, and Thailand [1,2,3]. The fruit, known as south wild jujube in China, with a light yellow flesh and sour–sweet taste, is valued for its substantial nutritional and therapeutic values in several regions of the world [4,5]. In Traditional Chinese Medicine (TCM), the fruit of *C. axillaris* has been historically utilized for its potential therapeutic properties to improve cardiac function as well as to regulate blood sugar levels and alleviate symptoms linked to diabetes [1,3,4,5,6,7]. Furthermore, it is used widely in China’s food industry for producing fruit pastilles and juice [8,9] and is utilized in Nepal for processing local products, e.g., pickles, fruit leathers, jam, and candy [1,10]. Regardless of insufficient reports on global sales data and socio-economic purposes, the market value of *C. axillaris* was found to be approximately USD 0.65 million in Kathmandu, Nepal [1,8,11]. *C. axillaris* as a functional food provides health benefits beyond basic nutrition due to its physiologically active components, which play an active role in promoting overall health and regulating disease progression [1,2,3,4,5,6,7,8,9,10].

So far, 26 compounds consisting of flavonoids, phenolic acids, organic acids, fatty acids, and sterols have been isolated from *C. axillaris* fruits [12,13,14,15], among which flavonoids represent a class of the most important and effective components that proven to have various biological activities, such as antioxidant [9,10,16], antiproliferative and cytotoxic [17,18], as well as anti-diabetic effects [19].

*α*-glucosidase is a kind of oligosaccharide hydrolase located in the small intestinal epithelial cell brush border responsible for the cleavage of *α*-1,4 bonds from the non-reducing end of *α*-linked oligosaccharide or *α*-glucan substrates to liberate glucose [20,21,22,23,24,25], which increases postprandial blood glucose, thereby causing obesity or diabetes [26]. Inhibiting the activity of *α*-glucosidase has been an effective therapeutic approach for hyperglycemia linked to type 2 diabetes mellitus. Rich in flavonoids which showed potent *α*-glucosidase inhibitory activity [27,28,29,30,31,32], *C. axillaris* fruit might be a potential source of hypoglycemic functional food.

Therefore, the present study was undertaken with the objective of isolating, identifying, and investigating the in vitro anti-diabetic potential of bioactive compounds from the fresh fruit of *C. axillaris*. Furthermore, aimed to reveal the possible inhibition mechanisms and binding mode of these compounds on *α*-glucosidase through molecular docking simulations.

## 2. Materials and Methods

### 2.1. General Procedure

UV spectra were measured on a Shimadzu UV-2401PC spectrophotometer (Shimadzu Co., Kyoto, Japan) with methanol as solvent. The optical rotations were recorded with a P-1020 polarimeter (JASCO, Tokyo, Japan). JASCO J-810 was used to obtain Experimental CD spectra, Bruker Av (600 MHz) spectrometers (Bruker Co., Karlsruhe, Germany) were used to measure 1D and 2D NMR spectra in methanol-*d4* (CD_3_OD). Chemical shifts were denoted by *δ* (ppm) and coupling constants were indicated as *J* (Hz), with TMS as an internal standard. IR spectra were measured using BioRad FtS-135 spectrophotometer (Bio-Rad, Richmond, CA, USA). Semi-preparative HPLC was performed on Waters 600 controller equipped with waters 2487 dual absorbance detector HPLC (Waters, Milford, USA) using COSMOSIL 5C_18_-MS-II (5 μm, 10 × 250 mm), Thermo scientific Hypersil GOLD aQ (5 μm, 10 × 250 mm) and Agilent ZORBAX SB-C18 (5 μm, 9.4 × 150 mm) packed columns. ESIMS and HRESIMS data were recorded using an Agilent 1290 UPLC/6540 Q-TOF instrument (Agilent Technologies, Palo Alto, CA, USA).

### 2.2. Chemicals and Reagents

LiChroprep Rp-18 (40–63 μm, Merck, Darmstadt, Germany), silica gel (200–300 mesh, Qingdao Haiyang Chemical Co. Ltd., Qingdao, China) and Sephadex LH-20 (25–100 μm, GE Healthcare Bio-Science AB, Uppsala, Sweden) were used as a stationary phase for column chromatography (CC). Precoated silica gel GF254 plates, 0.2–0.25 mm thick (Qingdao Haiyang Chemical Co., Qingdao, China), were used for thin layer chromatography (TLC) analysis, spots were visualized under a UV radiation by spraying with a sulfuric acid: ethanol (1:9, *v*/*v*) solution followed by heating. 4-Nitrophenyl-*α*-D-glucopyranoside (pNPG), *α*-glucosidase, quercetin and acarbose were purchased from sigma chemical (Merck KGaA, Darmstadt, Germany).

### 2.3. Materials

The fresh fruits of *Choerospondias axillaris* were collected in Kunming Botanical Garden, Kunming Institute of Botany (KIB), Chinese Academy of Sciences (CAS), Yunnan Province, China, on September 2020, and authenticated by Dr. En-De Liu from KIB, CAS. A voucher specimen (KIB-ZL-20200903) was stored in the State Key Laboratory of Phytochemistry and Plant Resources in West China, KIB, CAS.

### 2.4. Extraction and Isolation

The fresh fruits (95 kg) of *C. axillaris* were extracted with 80% aqueous acetone (4 × 80 L). The resulting solutions were filtered and concentrated under reduced pressure to give the acetone residue (5.79 kg). This was suspended into H_2_O and partitioned with EtOAc to obtain the EtOAc (732 g) extract and H_2_O residue. The EtOAc fraction (300 g) was subjected to CC using Sephadex LH-20 assisted with TLC, and gradient elution with MeOH/H_2_O (0:1–1:0, *v*/*v*) to give six fractions (Fr. 1–Fr. 6). Fr. 5 (54 g) underwent separation over silica gel CC, eluting with MeOH/CHCl_3_ (0:1–1:0, *v*/*v*) gradient system to obtain Fr. 5-1–Fr. 5-4. Fr. 5-4 was chromatographed with Sephadex LH-20 column, eluting with MeOH/H_2_O (0:1–1:0, *v*/*v*) and further purified with semi-preparative HPLC using Agilent ZORBAX SB-C18 (5 μm, 9.4 × 150 mm) packed column (CH_3_CN/H_2_O 16:84, *v*/*v*) to yield compounds **9** (25 mg, t_r_ = 18.9 min, CH_3_CN/H_2_O 17:83, *v*/*v*), **10** (15 mg, t_r_ = 20.9 min, CH_3_CN/H_2_O 16:84, *v*/*v*), **11** (18 mg, t_r_ = 20.9 min, CH_3_CN/H_2_O 20:80, *v*/*v*), **12** (22 mg, t_r_ = 40.7 min, CH_3_CN/H_2_O 20:80, *v*/*v*), **13** (20 mg, t_r_ = 23.4 min, CH_3_CN/H_2_O 28:72, *v*/*v*), **14** (10 mg, t_r_ = 12.2 min, CH_3_CN/H_2_O 13:87, *v*/*v*). Fr. 6 (43 g) was chromatographed over a silica gel column, employing MeOH/CHCl_3_ (0:1–1:0, *v*/*v*) gradient system as the eluent, to yield sub-fractions (Fr. 6-1–Fr. 6-5). Fr. 6-4 was fractionated on a column packed with RP-18 eluted with MeOH/H_2_O (0:1–1:0, *v*/*v*) to afford Fr. 6-4-1–Fr. 6-4-4. Fr. 6-4-4 was purified with semi-preparative HPLC using Thermo scientific Hypersil GOLD aQ (5 μm, 10 × 250 mm) packed column (CH_3_CN/H_2_O 82:18, *v*/*v*) to yield compounds **6** (6 mg, t_r_ = 50.5 min), **7** (5 mg, t_r_ = 35.3 min) and **8** (6.5 mg, t_r_ = 99 min). Fr. 6-5 was transferred to a column packed with silica gel, eluting with MeOH/CHCl_3_ (0:1–1:0, *v*/*v*), to afford sub-fractions Fr. 6-5-1–Fr. 6-5-4. Fr. 6-5-1 was repeatedly applied to CC over RP-18 with aqueous MeOH (40–100%) as eluent and further semi-preparative HPLC purification using Agilent ZORBAX SB-C18 (5 μm, 9.4 × 150 mm) packed column, yielded compounds **16** (8 mg, t_r_ = 11.3 min, CH_3_CN/H_2_O 17:83, *v*/*v*), **15** (5.3 mg, t_r_ = 16.05 min, CH_3_CN/H_2_O 17:83, *v*/*v*), **17** (7.4 mg, t_r_ = 30.2 min, CH_3_CN/H_2_O 17:83, *v*/*v*) and **18** (10 mg, t_r_ = 22.9 min, CH_3_CN/H_2_O 24:76, *v*/*v*). Silica gel CC separation of Fr. 6-5-1-6 eluting with MeOH/CHCl_3_ (1:9–2:8, *v*/*v*) and further semi-preparative HPLC purification using Thermo scientific Hypersil GOLD aQ column yield compounds **2** (7.3 mg, t_r_ = 21.8 min, CH_3_CN/H_2_O 79:21, *v*/*v*), **3** (3.3 mg, t_r_ = 18.2 min, CH_3_CN/H_2_O 81:19, *v*/*v*). Further purification of fraction Fr. 6-5-1-5 via semi-preparative HPLC applying COSMOSIL 5C_18_-MS-II (5 μm, 10 × 250 mm) packed column (CH_3_CN/H_2_O 80:20, *v*/*v*) yeilded compound **4** (8 mg, t_r_ = 36.0 min). Fr. 6-5-2 was successively subjected to RP-18 column, MeOH/H_2_O gradient elution of 50–100% and further resolved by semi-preparative HPLC employing a COSMOSIL 5C_18_-MS-II column (CH_3_CN/H_2_O 78:22, *v*/*v*) to afford compounds **1** (3.5 mg, t_r_ = 19.7 min) and **5** (4 mg, t_r_ = 23.5 min).

#### 2.4.1. Choerosponol F

Colorless solid, ${\left[\alpha \right]}_{D}^{20}$ 8.86 (*c* 0.07, MeOH); UV (MeOH) *λ*_max_ (log *ε*): 202.0 (3.34), 217.5 (2.97), 278.5 (2.13) nm; IR (KBr) *ν*_max_ 3404, 2926, 2854, 1061 cm^−1^; molecular formula: C_21_H_40_O_3_; HRESIMS *m*/*z* 385.2957 [M + HCOO]^−^ (calcd. for C_21_H_40_O_3_COOH^−^, 385.2959); ^1^H and ^13^C NMR (600 and 150 MHz, in CD_3_OD) spectroscopic data, presented in Table 1.

#### 2.4.2. Choerosponol G

Colorless oil, ${\left[\alpha \right]}_{D}^{20}$ 65.11 (*c* 0.09, MeOH); UV (MeOH) *λ*_max_ (log *ε*): 195.0 (3.16) nm; IR (KBr) *ν*_max_ 3414, 2924, 2853 cm^−1^; molecular formula: C_23_H_42_O_3_; HRESIMS *m*/*z* 389.3029 [M + Na]^+^ (calcd. for C_23_H_42_O_3_Na^+^, 389.3026); ^1^H and ^13^C NMR (600 and 150 MHz, in CD_3_OD) spectroscopic data, presented in Table 1.

#### 2.4.3. Choerosponol H

Colorless oil, ${\left[\alpha \right]}_{D}^{20}$ 49.60 (*c* 0.05, MeOH); UV (MeOH) *λ*_max_ (log *ε*): 195.0 (2.91) nm; IR (KBr) *ν*_max_ 3405, 2923, 2853 cm^−1^; molecular formula: C_25_H_46_O_3_; HRESIMS *m*/*z* 417.3337 [M + Na]^+^ (calcd. for C_25_H_46_O_3_Na^+^, 417.3339); ^1^H and ^13^C NMR (600 and 150 MHz, in CD_3_OD) spectroscopic data, presented in Table 1.

#### 2.4.4. Choerosponol I

Colorless oil, ${\left[\alpha \right]}_{D}^{20}$ 19.45 (*c* 0.11, MeOH); UV (MeOH) *λ*_max_ (log *ε*): 195.0 (3.04), 234.0 (2.97) nm; IR (KBr) *ν*_max_ 3394, 2924, 2853, 1658 cm^−1^; molecular formula: C_23_H_40_O_3_; HRESIMS *m*/*z* 387.2868 [M + Na]^+^ (calcd. for C_23_H_40_O_3_Na^+^, 387.2870); ^1^H and ^13^C NMR (600 and 150 MHz, in CD_3_OD) spectroscopic data, presented in Table 2.

#### 2.4.5. Choerosponol J

Colorless amorphous solid, ${\left[\alpha \right]}_{D}^{20}$ 41.60 (*c* 0.05, MeOH); UV (MeOH) *λ*_max_ (log *ε*): 195.0 (2.95), 234.0 (2.89) nm; IR (KBr) *ν*_max_ 3406, 2925, 2853, 1661 cm^−1^; molecular formula: C_23_H_40_O_4_; HRESIMS *m*/*z* 425.2909 [M + HCOO]^−^ (calcd. for C_23_H_40_O_4_COOH^−^, 425.2909). ^1^H and ^13^C NMR (600 and 150 MHz, in CD_3_OD) spectroscopic data, presented in Table 2.

#### 2.4.6. Choerosponol K

Colorless amorphous solid, ${\left[\alpha \right]}_{D}^{20}$ 7.17 (*c* 0.12, MeOH); UV (MeOH) *λ*_max_ (log *ε*): 204.0 (4.37), 251.0 (4.03), 294.0 (3.56) nm; IR (KBr) *ν*_max_ 3413, 2922, 2851, 1605, 1468, 1198 cm^−1^; molecular formula: C_23_H_34_O_2_; HRESIMS *m*/*z* 341.2479 [M − H]^−^ (calcd. for C_23_H_34_O_2_^−^ 341.2486); ^1^H and ^13^C NMR (600 and 150 MHz, in CD_3_OD) spectroscopic data, presented in Table 2.

#### 2.4.7. Choerosponol L

Colorless oil, ${\left[\alpha \right]}_{D}^{20}$ 6.00 (*c* 0.05, MeOH); UV (MeOH) *λ*_max_ (log *ε*): 205.0 (4.27), 251.0 (3.98), 294.0 (3.52) nm; IR (KBr) *ν*_max_ 3427, 2924, 2853, 1607, 1468, 1197 cm^−1^; molecular formula: C_21_H_30_O_2_; HRESIMS *m*/*z* 337.2137 [M + Na]^+^ (calcd. for C_21_H_30_O_2_Na^+^ 337.2138); ^1^H and ^13^C NMR (600 and 150 MHz, in CD_3_OD) spectroscopic data, presented in Table 2.

### 2.5. Quantum Chemistry ECD Calculation

Conformational analysis was performed with the MMFF forcefield (Merck Molecular forcefield) using the Monte Carlo algorithm implemented in Spartan’10 software (Wavefunction, Irvine, CA, USA, 2018). ECD calculations were carried out, following optimizing the conformers at B3LYP/6-31+g (d, p) level in MeOH using the CPCM polarizable conductor calculation model, the conformers with a Boltzmann population of over 5% were chosen. The theoretical calculation of ECD was conducted in MeOH using time-dependent density functional theory (TD-DFT) at the B3LYP/6-311+g (d, p) level for all conformers of compound 18b. Rotatory strengths for a total of 50 excited states were calculated. ECD spectra were generated using the program. SpecDis 1.6 (University of Würzburg, Würzburg, Germany) and GraphPad Prism 5 (University of California, San Diego, CA, USA) from dipole-length rotational strengths by applying Gaussian band shapes with sigma = 0.3 eV.

### 2.6. α-Glucosidase Inhibitory Assay

The *α*-glucosidase inhibitory activity was evaluated according to a previous method [33]. In brief, *α*-glucosidase solution (0.025 U/mL, dissolved in 0.1 M phosphate-buffered saline (PBS) solution with a pH of 6.8), was mixed with 50 μM inhibitor and incubated at 37 °C for 10 min. pNPG solution (1 μM, dissolved in 0.1 M PBS solution) was added to the mixed solution and incubated at 37 °C for 40 min. Sodium carbonate (1 M) solution was added to terminate the reaction. Quercetin and acarbose were used as a positive control. Quercetin is selected as a reference in *α*-glucosidase assay because of its efficacy as a therapeutic inhibitor; it binds strongly within the binding region (active site) of the protein and forms a compact structure, whereas acarbose is a common medication utilized in clinical practice to manage postprandial hyperglycemia [30,34,35]. The absorbance was recorded at 405 nm usin a microplate reader. The *α*-glucosidase inhibitory rate was calculated as follows:

Inhibitory rate% = [(A0 − A1)/A0)] × 100%, where:

A0 and A1 represent the absorbance of the blank group (containing enzyme, pNPG, PBS buffer, and DMSO), the sample group (containing enzyme, pNPG, PBS buffer, and samples), respectively. IC_50_ values were computed based on the Reed–Muench method [36].

### 2.7. Molecular Docking Analysis

The X-ray crystallographic structures of *α*-glucosidase from *Saccharomyces cerevisiae* (PDB ID: 3A4A) at a resolution of 1.6 Å were downloaded from the RCSB Protein Data Bank (https://www.rcsb.org (accessed on 7 February 2024)); prior to docking, water molecules and hetero atoms of the protein were removed, missing atoms were repaired, polar hydrogens and kollman charges were added. The 3D structure of the ligands was prepared in Chem3D by energy minimization using the MM2 force field, saved in Sybyl Mol2 format and converted to AutoDock PDBQT format with Open Babel 3.1.1 chemical toolbox [37]. Molecular docking simulations were performed using AutoDockTools-1.5.7 with Lamarckian genetic algorithm 4.2 to understand the interactions and binding of isolated compounds with selected proteins following previously described protocols with minor modification [38,39]. The docking results were visualized and analyzed using the PyMOL molecular graphics system and open-source visualization tools [40,41,42].

### 2.8. Statistical Analysis

All data were analyzed and presented as mean values ± SD from three independent assays (*n* = 3). The IC_50_ values were determined by Graph Pad Prism version 5 (University of California, San Diego, CA, USA).

## 3. Results and Discussion

### 3.1. Structural Elucidation

Eighteen compounds (**1**–**18**) (Figure 1) were isolated from the 80% aqueous acetone extract of C. axillaris fresh fruits by repeated CC over Sephadex LH-20, RP-18, and silica gel, followed by semi-preparative HPLC purification. Seven of them are new (**1**–**7**), while the known ones **8**–**18** were identified by comparison of their spectroscopic data with those reported in the literature, as one benzofuran derivative, choerosponol A (**8**) [18], six flavonoids, catechin (**9**) [43], dihydroquercetin (**10**) [44], dihydrokaempferol (**11**) [44], eriodictyol (**12**) [45], naringenin (**13**) [46], quercetin-3-O-*α*-L-rhamnoside (**14**) [47], and four lignans, (+)-isolariciresinol (**15**) [48], (−)-lyoniresinol (**16**) [49], (−)-secoisolariciresinol (**17**) [50], and lirioresinol (**18**) [51]. The known lignans **15**–**18** were isolated from the genus Choerospondias for the first time.

Compound **1** was obtained as a colorless solid. Its molecular formula was established as C_21_H_40_O_3_ based on HRESIMS data, which provided a [M + HCOO]^−^ ion at *m*/*z* 385.2957 (calcd. 385.2959). The IR band at 3404 cm^−1^ revealed the presence of a hydroxyl group. The ^1^H NMR and ^13^C NMR spectra of **1** displayed the presence of two oxymethine protons resonating at *δ*_H_ 3.84 (1H, tt, *J* = 11.1, 4.4 Hz) and *δ*_H_ 3.36 (1H, dd, *J* = 10.7, 5.4 Hz) with the corresponding carbon resonances at *δ*_C_ 67.4 (C-4) and *δ*_C_ 73.9 (C-1), respectively, in HSQC experiment and a quaternary oxygenated carbon at *δ*_C_ 75.4 (C-2). Moreover, proton signals ascribed to methylene groups at *δ*_H_ 1.95 (1H, ddd, *J* = 13.3, 4.4, 2.8 Hz, H-3a), 1.90 (1H, ddd, *J* = 12.5, 4.4, 2.9 Hz, H-5a), 1.70 (2H, m, H-6) and overlapping peaks at *δ*_H_ 1.23 (2H, m, H-3b and H-5b) along with carbon shifts at *δ*_C_ 43.7 (C-3), 29.1 (C-6) and 34.3 (C-5) were observed (Table 1). The 1,2,4-trihydroxysubstituted cyclohexane ring were established by ^1^H-^1^H COSY correlations between H-3/H-4, H-1/H-6, H-5/H-6 and H-4/H-5 as well as, HMBC correlations of H-4 (*δ*_H_ 3.84) to C-6 and C-2, H-1 (*δ*_H_ 3.36) to C-1′ (*δ*_C_ 40.7) and C-5. The location of Δ^10′,11′^ double bond was deduced from the HMBC correlation of terminal methyl proton H-15′ (*δ*_H_ 0.92) to C-13′ (*δ*_C_ 33.1), overlapping vinylic protons H-10′/H-11′ to C-8′ (*δ*_C_ 30.3) and to C-13′, and protons H-9′/H-12′ (*δ*_H_ 2.04) to C-7′ (*δ*_C_ 30.6) and C-14′ (*δ*_C_ 23.4), respectively. The assignment of the (*Z*)-stereochemistry to the side-chain double bond in the alkenyl chain was established by comparison of the chemical shifts observed in the ^13^C NMR for C-9′ and C-12′ with the literature data, instead of a higher chemical shift value of *δ*_C_ 32.6 exhibited for the (*E*)-configuration; compound **1** possesses chemical shift of C-9′ (*δ*_C_ 28.1) and C-12′ (*δ*_C_ 27.9) [52,53,54,55,56]. The ROESY cross-peak correlation observed between H-1/H-1′, H-4/H-5, and H-5/H-6 suggested OH-1, and OH-2 to be β-oriented and OH-4 to be α-oriented. The experimental ECD spectra aligned with the calculated ECD spectra bearing a 1*S*,2*R*,4*S* configuration (Figure 1). Thus, the absolute configurations were assigned, the structure of **1** was established as (1*S*,2*R*,4*S*)-2-[10′(*Z*)-pentadecenyl]- cyclohexane-1,2,4-triol and named as choerosponol F.

Based on HRESIMS analysis, compound **2** was determined to have a molecular formula of C_23_H_42_O_3_, with a molecular ion peak at *m*/*z* 389.3029 [M + Na]^+^ (calcd. for 389.3026). The ^1^H NMR (Table 1) spectrum showed a pair of olefinic protons resonances at *δ*_H_ 5.75 (1H, dd, *J* = 10.2, 2.0 Hz) and 5.54 (1H, dt, *J* = 10.2, 2.6 Hz), two oxymethine proton resonates at *δ*_H_ 4.39 (1H, ddd, *J* = 9.5, 5.5, 2.0 Hz), and 4.00 (1H, d, *J* = 2.6 Hz) together with methylene protons at *δ*_H_ 2.17 (1H, dd, *J* = 13.2, 5.5 Hz, H-3a) and *δ*_H_ 1.42 (1H, dd, *J* = 13.2, 9.5 Hz, H-3b). The ^13^C NMR data coupled with the HSQC spectra showed that the cyclohexene ring is substituted with three hydroxy groups, at C-4 (*δ*_C_ 66.1), C-2 (*δ*_C_ 75.1) and C-1 (*δ*_C_ 70.8) along with olefinic carbon shifts at C-5 (*δ*_C_ 133.3) and C-6 (*δ*_C_ 130.7) (Table 1), which is in consistence with correlations observed (as depicted in Figure 2) in the HMBC spectrum from H-4 (*δ*_H_ 4.39) to C-6, H-3a (*δ*_H_ 1.42)/H-3b (*δ*_H_ 2.17) to C-5, C-1 and C-1′ (*δ*_C_ 40.4), and from olefinic protons H-5 (*δ*_H_ 5.75) to C-1 and C-3, H-6 (*δ*_H_ 5.54) to C-4 and C-2. In comparison with the 1D/2D NMR spectra in the reported literature, compound **2** was similar to the previously identified compound from *Lannea schimperi* (Anacardiaceae) [57]; the only differences in the structure were due to the location and geometry of the double bond within the alkenyl chain, which were supported by the ^1^H-^1^H COSY correlations of H-11′/H-12′, H-13′/H-14′ and HMBC correlation of terminal methyl proton H-17′ (*δ*_H_ 0.92) to C-15′ (*δ*_C_ 33.1), overlapping vinylic protons H-12′/H-13′ (*δ*_H_ 5.35) to C-10′ (*δ*_C_ 30.9) and to C-15′ (*δ*_C_ 33.1), permitted the position of the double bond Δ^12′,13′^. The experimental ECD spectrum of **2** matched the calculated ECD curve of **1**; consequently, (1*S*,2*R*,4*S*) absolute configuration was established for choerosponol G. Therefore, compound **2** was identified as (1*S*,2*R*,4*S*)-2- [12′(*Z*)-heptadecenyl]-cyclohex-5-en-1,2,4-triol.

Compound **3** exhibited a [M + Na]^+^ molecular ion peak at *m/z* 417.3337 (calcd. for 417.3339) in the HRESIMS experiment, verifying the molecular formula as C_25_H_46_O_3_. The ^13^C NMR and HSQC experiment of compound **3** revealed the presence of 25 carbon resonances attributed to one methyl (*δ*_C_ 14.4), 17 methylenes (*δ*_C_ 41.5–23.4), 4 methines (*δ*_C_ 133.3, 130.8, 130.8 and 130.8), 2 oxygenated methines (*δ*_C_ 66.2 and 70.9) and 1 quaternary oxygenated carbon (*δ*_C_ 75.2). Proton shifts at *δ*_H_ 5.74 (1H, dd, *J* = 10.2, 2.0 Hz) and 5.53 (1H, dt, *J* = 10.2, 2.5 Hz) in the ^1^H NMR spectrum (Table 1) deduce a pair of olefinic protons with corresponding carbon signals in the ^13^C NMR that were detected at *δ*_C_ 133.3 and 130.8, respectively. Two oxymethine protons resonating at *δ*_H_ 4.38 (1H, ddd, *J* = 9.5, 3.9, 2.0 Hz) and 4.00 (1H, d, *J* = 2.5 Hz), methylene protons at *δ*_H_ 2.15 (1H, dd, *J* = 13.2, 3.9 Hz) and *δ*_H_ 1.42 (1H, dd, *J* = 13.2, 9.5 Hz) were also observed. The location of three hydroxyl groups on the cyclohexene moiety was determined from the correlations observed in the HMBC spectrum from oxymethine proton H-4 (*δ*_H_ 4.38) to C-6 (*δ*_C_ 130.8) and C-2 (*δ*_C_ 75.2), methylene proton signals H-3a (*δ*_H_ 1.42) and H-3b (*δ*_H_ 2.15) to C-5 (*δ*_C_ 133.3), C-1 (*δ*_C_ 70.9) and C-1′ (*δ*_C_ 40.4), and from olefinic protons H-5 (*δ*_H_ 5.74) to C-3 (*δ*_C_ 41.5) and C-1, H-6 (*δ*_H_ 5.53) to C-4 (*δ*_C_ 66.2) and C-2. ROESY correlations observed between H-1/H-1′, H-1/H-6, and H-4/H-5 suggested the cis orientation of OH-1 and OH-2; subsequently, from the established absolute configuration of compound **1**, the absolute configuration of compound **3** was determined as (1*S*,2*R*,4*S*). The structure of **3** was therefore established as (1*S*,2*R*,4*S*)-2-[14′(*Z*)-nonadecenyl]-cyclohex-5-en-1,2,4-triol and given the trivial name choerosponol H.

HRESIMS data of **4** provided a [M + Na]^+^ ion peak at *m*/*z* 387.2868 (calcd. for 387.2870) confirming its molecular formula as C_23_H_40_O_3_. The IR spectrum showed a broad OH absorption band at 3394 cm^−1^ and the absorption of a carbonyl group at 1658 cm^−1^, indicating an *α*,*β*-unsaturated ketone. The ^1^H NMR spectrum (Table 2) showed olefinic proton resonance at *δ*_H_ 5.88 (1H, s), a triplet signal of vinaylic protons at *δ*_H_ 5.34 (2H, t, *J* = 5.6 Hz), two oxymethine proton resonates at *δ*_H_ 4.39 (1H, dd, *J* = 8.2, 4.6 Hz) and 3.86 (1H, hept, *J* = 4.5 Hz). The ^13^C NMR and HSQC spectra of the *α*,*β*-unsaturated cyclohexanone ring displayed chemical shift value of *δ*_C_ 201.7 ascribed for C=O, monohydroxy substituted at C-4 (*δ*_C_ 68.7)/*δ*_H_ 4.39, besides exhibited olefinic carbon shifts at C-2 (*δ*_C_ 128.0)/*δ*_H_ 5.88 and C-3 (*δ*_C_ 167.6), whereby their positions on the cyclohexanone ring were established via ^1^H-^1^H COSY correlations between H-4/H-5 and H-5/H-6 as well as the HMBC correlation of H-2 to C-6 (*δ*_C_ 35.5), C-4, and C-1′ (*δ*_C_ 43.0), H-4 to C-2, C-6 and C-1′, H-6a/H-6b (*δ*_H_ 2.37, 2.54) to C-1 and C-4, H-1′a/H-1′b (*δ*_H_ 2.44, 2.59) to C-3′ (*δ*_C_ 38.7), C-4, and C-2, H-5a/H-5b (*δ*_H_ 1.97, 2.23) to C-1 and C-3. The calculated ECD spectrum of **4** matched well with the experimental ECD curve of **4** (Figure 3)**,** permitting the assignment of the (4*R*,2′*R*) absolute configuration; hence, compound **4** determined as (4*R*,2′*R*)-dihydroxy-3-[12′(*Z*)-heptadecenyl]-2-cyclohexenone given the trivial name choerosponol I.

Compound **5** was acquired as a colorless amorphous solid, and its molecular formula C_23_H_40_O_4_ was determined based on its HRESIMS data, which provided a [M + HCOO]^−^ ion at *m*/*z* 425.2909 (calcd. for 425.2909). The ^1^H and ^13^C NMR data for **5** (Table 2) were comparable to those obtained for **4**; however, signals for an additional oxygenated methine *δ*_C_ 71.0 (C-5) group were observed. Its ^1^H NMR and ^13^C NMR spectrum indicated the presence of an *α*,*β*-unsaturated carbonyl group *δ*_C_ 199.9 (C-1), olefinic proton at *δ*_H_ 5.91 (1H, s)/*δ*_C_ 128.4 (C-2), *δ*_C_ 164.8 (C-3), three oxymethine protons resonating at *δ*_H_ 4.17 (1H, dd, *J* = 6.8, 1.4 Hz), 3.95 (1H, ddd, *J* = 9.5, 6.8, 4.3 Hz) and 3.87 (1H, hept, *J* = 4.5 Hz) with the corresponding carbon resonance at *δ*_C_ 74.4, *δ*_C_ 71.0 and *δ*_C_ 72.7, respectively, showed in the HSQC experiment. The exact location in the cyclohexanone ring was feature from ^1^H-^1^H COSY correlation between H-4/H-5, H-5/H-6, H-1′/H-2′, H-2′/H-3′ and the HMBC spectrum correlation of H-2 (*δ*_H_ 5.91) to C-6 (*δ*_C_ 44.4), C-1′ (*δ*_C_ 42.8), C-4 (*δ*_C_ 74.4), H-4 (*δ*_H_ 4.17) to C-6, C-1′, H-5 (*δ*_H_ 3.95) to C-3, C-1 (*δ*_C_ 199.9), and nonequivalent methylene protons H-6a/6b (*δ*_H_ 2.42, 2.73) to C-4, C-2, H-1′ (*δ*_H_ 2.46, 2.61) to C-3′ (*δ*_C_ 38.5), C-4, C-2. The calculated ECD spectrum of (4*S*,5*S*,2′*R*)-**5** matched the experimental ECD curve (Figure 3). Consequently, the absolute configuration of **5** was determined as (4*S*,5*S*,2′*R*)-trihydroxy-3-[12′(*Z)*-heptadecenyl]-2-cyclohexenone given the trivial name choerosponol J.

The molecular formula of **6** was assigned as C_23_H_34_O_2_, based on HRESIMS data revealing a [M − H]^−^ ion at *m*/*z* 341.2479 (calcd 341.2486). The ^1^H NMR (Table 2) spectrum showed aromatic protons at *δ*_H_ 7.15 (1H, d, *J* = 8.7 Hz), 6.84 (1H, d, *J* = 2.5 Hz), 6.66 (1H, dd, *J* = 8.7, 2.5 Hz) and 6.25 (1H, s). The HMBC correlation signals from H-7 (*δ*_H_ 7.15) to C-5 (*δ*_C_ 153.9) and C-3′ (*δ*_C_ 131.1), H-6 (*δ*_H_ 6.66) to C-4 (*δ*_C_ 106.1) and C-7′ (150.6), H-4 (*δ*_H_ 6.84) to C-7′, C-6 (112.5) and long-range correlations H-3 (*δ*_H_ 6.25) to C-4, C-7′, H-1′′ (*δ*_H_ 2.67) to C-3 (*δ*_C_ 102.8), C-4, H-2′′ (*δ*_H_ 1.68) to C-2 (*δ*_C_ 161.4), besides the ^1^H-^1^H COSY correlation between aromatic protons H-6/H-7, and H-1′′/H-2′′, reveal the benzofuran arene moiety. Signals referable to methylene protons at *δ*_H_ 2.67 (2H, t, *J* = 7.6 Hz), 1.68 (2H, p, *J* = 7.4 Hz), a series of overlapping peaks at *δ*_H_ 1.24–1.35 (16H, m), olefinic proton resonance at 5.32 (2H, m) and terminal methyl proton at *δ*_H_ 0.89 (3H, t, *J* = 7.0 Hz) indicated the presence of a long alkenyl chain in **6**. ^1^H and ^13^C NMR data of **6** were closely related to those of **8** [18]. However, compound **6** was missing signals corresponding to two methylene carbons occurring in the unsaturated hydrocarbon chain which was confirmed by the HRESIMS experiment. Therefore, the structure of compound **6** was established (Figure 1) which is attributed to be 2-[10′(*Z*)-pentadecenyl]-benzofuran-5-ol and named as choerosponol K.

HRESIMS data of **7** revealed a [M + Na]^+^ ion at *m*/*z* 337.2137 (calcd. for C_21_H_30_O_2_Na^+^ 337.2138) indicating the molecular formula of C_21_H_30_O_2_. The ^13^C-NMR spectra (Table 2) displayed 21 carbon resonances, attributed to one methyl, ten methylenes, six methines (two olefinic, four aromatic), and four quaternary carbons (one carbonyl oxygenated). Aromatic protons at *δ*_H_ 7.16 (1H, d, *J* = 8.7 Hz), 6.83 (1H, d, *J* = 2.5 Hz), 6.65 (1H, dd, *J* = 8.7, 2.5 Hz) and 6.27 (1H, s) were observed in the ^1^H NMR (Table 2) spectrum. The presence of benzofuran arene moiety was confirmed by the HMBC correlations from H-7 (*δ*_H_ 7.16) to C-5 (*δ*_C_ 153.9) and C-3′ (*δ*_C_ 131.1), H-6 (*δ*_H_ 6.65) to C-4 (*δ*_C_ 106.1) and C-7′ (150.6), H-4 (*δ*_H_ 6.83) to C-7′, C-6 (112.5) and H-3 (*δ*_H_ 6.27) to C-4, C-7′, H-1′′ (*δ*_H_ 2.69) to C-3 (*δ*_C_ 102.8), C-4, H-2′′ (*δ*_H_ 1.70) to C-2 (*δ*_C_161.5). The 1D and 2D NMR data for **7** were comparable to those obtained for compound **6**, which showed that the structure of the cyclic moiety was identical, and the differences consisted of the absence of two methylene groups in the alkenyl chain; these observations were supported by HRESIMS data of **7** [M + Na]^+^ ion at *m*/*z* 337.2137, indicating a molecular formula of C_21_H_30_O_2_. Hence, compound **7** was established (Figure 1) as 2-[8′(*Z*)-tridecenyl]-benzofuran-5-ol and named as choerosponol L.

### 3.2. α-Glucosidase Inhibition

The *α*-glucosidase inhibitory activity of compounds isolated from *C. axillaris* was evaluated. As shown in Table 3, most compounds were demonstrated with potential *α*-glucosidase inhibitory activity. Benzofuran derivatives **6**, **7**, and **8** exert superior inhibitory activity than quercetin and acarbose, with IC_50_ values of 3.5 ± 0.12, 2.26 ± 0.06, 4.5 ± 0.10 μM, respectively. Out of the tested flavonoids, **9**, **12**, and **13** were proved to demonstrate excellent *α*-glucosidase potency with IC_50_ values of 43.9 ± 0.96, 38.3 ± 0.53, 36.3 ± 1.3, respectively. Compounds **10**, **11** and **14**, exhibit comparable *α*-glucosidase inhibitory, whereas weak activity was noted for lignans **15**, and **17**. At a concentration of 50 μM, compounds **1**–**5** showed no inhibitory effects on *α*-glucosidase.

### 3.3. Molecular Docking Studies

To rationalize the significant in vitro *α*-glucosidase inhibition results achieved, molecular docking studies were designed [58,59,60,61], thereby illustrating the docking affinities and the docking conformations of the isolated compounds with *α*-glucosidase from *Saccharomyces cerevisiae* isomaltase (PDB ID: 3A4A), a principal diabetes-related enzyme located in the small intestine, which catalyzes the cleavage of *α*-glucopyranoside bond in oligosaccharides and disaccharides, leading to increases in blood glucose concentration.

The molecular docking results, including the binding affinities of ligands (**6**, **7**, **8** and quercetin) that exhibited potent *α*-glucosidase inhibition and their interactions (hydrogen bonds and hydrophobic interactions), are shown in Figure 4. The highest docking score indicates greater affinity of the compound towards the protein which finds well fit inside the binding pocket of the protein. This is consistent with the results of the *α*-glucosidase inhibitory assay.

Compound **6** demonstrated the highest binding affinity of −9.77 kcal/mol which is attributed to the hydrogen bonding interactions observed with Arg 442A (2.8 Å), Asp 215A (2.8 Å), and His112A (3.3 Å), as well as established hydrophobic bond interactions with Arg 315A (3.8 Å), Phe 178A (3.2 Å), Val 216A (3.8 Å), Tyr 158A (3.4 Å), Lys 156A (3.2 Å), Tyr 72A (3.1 Å), and Phe 314A (3.4 Å), implying that the hydroxyl group and the pi-electrons on the arene moiety may play an important role in the inhibition of the *α*-glucosidase and justify its interesting in vitro inhibition rate.

Compound **7** scored enhanced docking score of −9.49 kcal/mol where amino acid residues Arg 442A (2.3 Å), Asp 215A (2.8 Å), and His112A (2.9 Å), forming hydrogen bond interactions with benzofuran ring, and interactions with Lys 156A (3.6 Å), Arg 315A (3.7 Å), Tyr 158A (3.2 Å), Tyr 316A (3.7 Å), Val 216A (3.3 Å), Tyr 72A (3.3 Å), Phe 159A (3.8 Å), Phe 178A (3.8 Å) and Phe 314A (3.5 Å), residues through hydrophobic interaction, were also illustrated. Evidently, this compound exhibited potent *α*-glucosidase inhibition, which translated into analogous in vitro IC_50_ values of 2.26 ± 0.06 μM.

Compound **8** exhibits excellent binding affinities of −9.62 kcal/mol, and hydrogen bond interactions were executed between the hydroxyl group and the key amino acid residue Arg 442A (2.5 Å), Asp 215A (2.5 Å), and His112A (2.2 Å). The interactions with Tyr 72A (3.1 Å), Lys 156A (3.9 Å), Val 216A (3.1 Å), Tyr 158A (3.3 Å), Phe 159A (3.7 Å), Leu 177A (3.7 Å), Phe 178A (3.8 Å), Phe 178A (3.2 Å), Phe 314A (3.6 Å) and Arg 315A (3.2 Å) residues through hydrophobic interaction were also demonstrated, which are extrapolated to its impressive analogous in vitro IC_50_ values.

The positive control quercetin showed a binding affinity of −7.73 kcal/mol, which is consistent with the previous study by Qin et al. [28], who proved the reason why it possesses less *α*-glucosidase inhibition compared to compounds **6**, **7** and **8**. Notably, it formed a hydrogen bond interaction with amino acid residues Asp 352A, Arg 315A, Asp 215A and Gln 279.

### 3.4. Structural Activity Relationship

The new compounds possess analogous alkenyl side chains; nonetheless, the inhibitory effect significantly changed when a heterocyclic aromatic ring was attached instead of a non-aromatic cyclic ring. The results showed that the compounds with benzofuran arene moiety tend to exert remarkable *α*-glucosidase inhibitory activity than alkenyl cyclohexenol and cyclohexenone derivatives. These significant differences in activity indicate that the presence of a heterocyclic aromatic ring undoubtedly plays a crucial role in the compounds exerting *α*-glucosidase inhibitory activity, which is attributed to the high docking score and hydrogen, hydrophobic interactions in the docking study.

The flavanonols dihydroquercetin (**10**), dihydrokaempferol (**11**) and flavanones eriodictyol (**12**) and naringenin (**13**) possess carbonyl groups; however, they differ in the number of OH groups, resulting in the observed differences in inhibition of *α*-glucosidase. Flavan-3-ol (**9**), with the absence of the carbonyl group, displayed a notable *α*-glucosidase inhibition; furthermore, the presence of the rhamnosyl group at C-3 on quercitrin (**14**) altered the enzyme inhibition activity, which is consistent with previous research findings by Li et al., [29].

Lignans (+)-isolariciresinol (**15**), (−)-lyoniresinol (**16**), (−)-secoisolariciresinol (**17**) and lirioresinol (**18**) displayed various weak inhibitory activity; a methoxy group in the aromatic ring could be responsible for the differences observed in *α*-glucosidase inhibition.

## 4. Conclusions

In this study, chemical investigation of the fresh fruits of *C. axillaris* led to the isolation and structural elucidation of 18 compounds including 7 new (**1**–**7**) and 11 known (**8**–**18**), comprised of 5 alkenyl (cyclohexenols and cyclohexenones) derivatives (**1**–**5**), 3 benzofuran derivatives (**6**–**8**), 6 known flavonoids (**9**–**14**), and 4 known lignans (**15**–**18**). Compounds **15**–**18** were isolated from the genus *Choerospondias* for the first time. Compounds **6**–**9**, **12**, and **13** exerted strong *α*-glucosidase inhibitory activity with IC_50_ values from 2.26 to 43.9 μM. The results of molecular docking simulations illustrate good coherence with in vitro *α*-glucosidase inhibition. Interestingly, benzofuran derivatives were proved to have potential hypoglycemic activity, suggesting these compounds may also play a substantial role in the hypoglycemic activity of *C. axillaris* fruit. The results suggested that *C. axillaris* fruit could be an ideal source of hypoglycemic functional foods.

## Figures and Tables

**Figure 1 foods-13-01495-f001:**
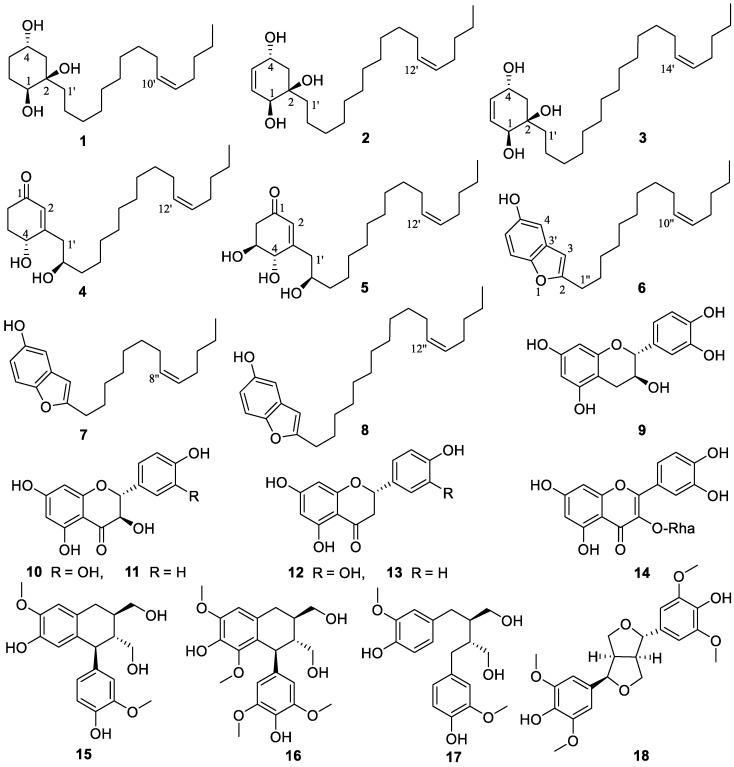
Compounds **1**–**18** isolated from the fruits of *Choerospondias axillaris*.

**Figure 2 foods-13-01495-f002:**
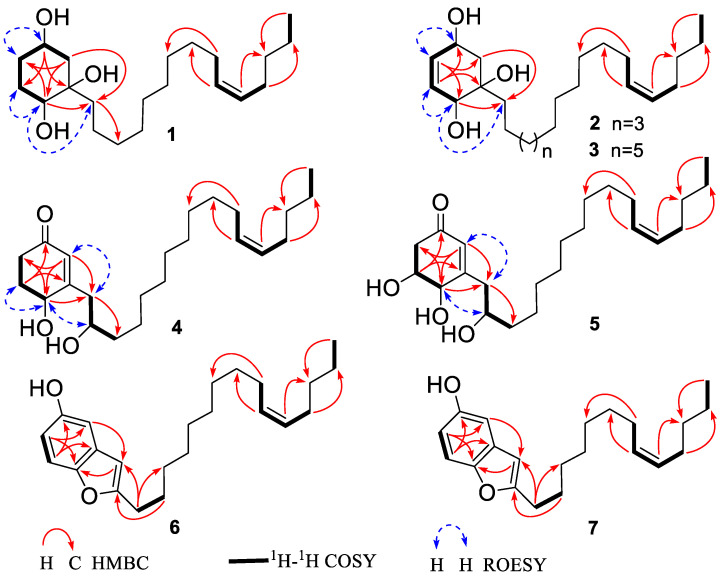
Key HMBC, ^1^H-^1^H COSY and ROESY correlations of compounds **1**–**7**.

**Figure 3 foods-13-01495-f003:**
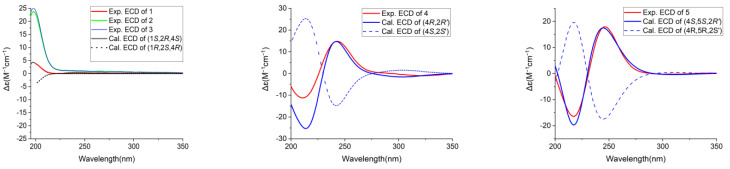
Calculated and experimental ECD curves of compounds **1**–**5**.

**Figure 4 foods-13-01495-f004:**
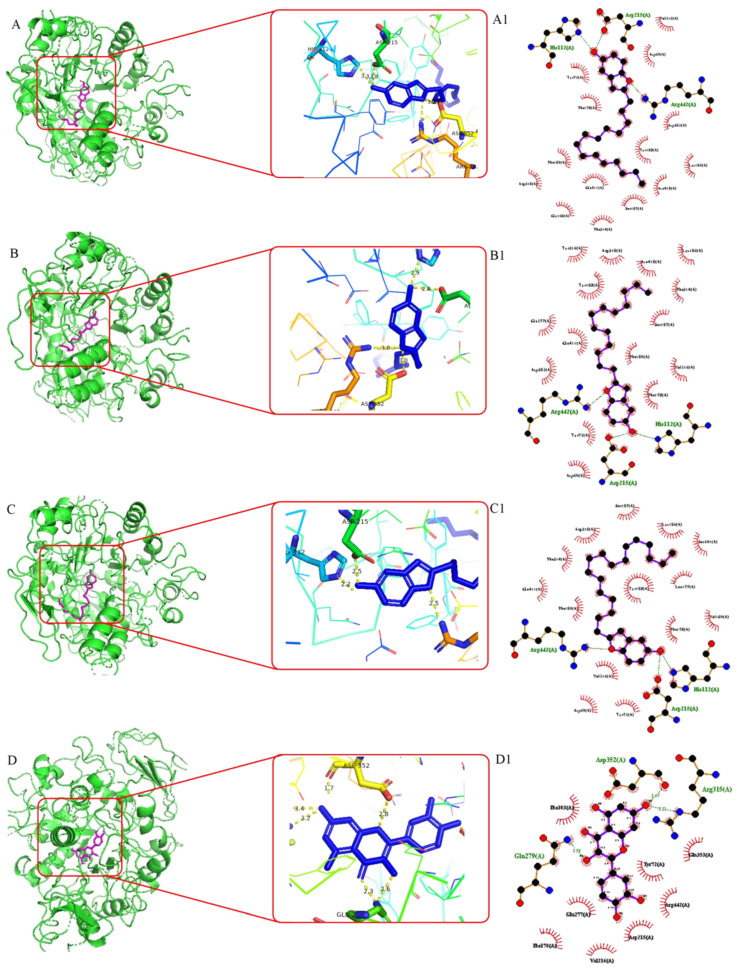
Molecular docking study of compounds binding to *α*-glucosidase protein. (**A**–**D**): 3D ligand interaction diagram of **6**, **7**, **8** and quercetin with *α*-glucosidase and a close view of the binding active site with key residues that interacted to ligands (right); (**A1**–**D1**): 2D protein–ligand interaction diagram of **6**, **7**, **8**, quercetin and *α*-glucosidase complex.

**Table 1 foods-13-01495-t001:** ^13^C (150 MHz) and ^1^H (600 MHz) NMR spectroscopic data of compounds **1**–**3** in CD_3_OD (*δ* in ppm, *J* in Hz).

No.	1		No.	2		No.	3	
	*δ_C_*, Type	*δ_H_* (*J* in Hz)		*δ_C_*, Type	*δ_H_* (*J* in Hz)		*δ_C_*, Type	*δ_H_* (*J* in Hz)
1	73.9	3.36 (dd, 10.7, 5.4)	1	70.8	4.00 (d, 2.6)	1	70.9	4.00 (d, 2.5)
2	75.4		2	75.1		2	75.2	
3	43.7	1.23 (dd, 13.3, 11.1) ^a^ 1.95 (ddd, 13.3, 4.4, 2.8)	3	41.4	1.42 (dd, 13.2, 9.5) 2.17 (dd, 13.2, 5.5)	3	41.5	1.42 (dd, 13.2, 9.5) 2.15 (dd, 13.2, 3.9)
4	67.4	3.84 (tt, 11.1, 4.4)	4	66.1	4.39 (ddd, 9.5, 5.5, 2.0)	4	66.2	4.38 (ddd, 9.5, 3.9, 2.0)
5	34.3	1.23 (m) ^a^ 1.90 (ddd, 12.5, 4.4, 2.9)	5	133.3	5.75 (dd, 10.2, 2.0)	5	133.3	5.74 (dd, 10.2, 2.0)
6	29.1	1.70 (m)	6	130.7	5.54 (dt, 10.2, 2.6)	6	130.8	5.53 (dt, 10.2, 2.5)
1′	40.7	1.51 (m) 1.58 (m)	1′	40.4	1.60 (m)	1′	40.4	1.59 (m)
2′	24.5	1.29–1.36 (m) ^a^	2′	24.7	1.28–1.39 (m) ^a^	2′	24.7	1.28–1.37 (m) ^a^
3′	31.5	1.29–1.36 (m) ^a^	3′	31.5	1.28–1.39 (m) ^a^	3′	31.5	1.28–1.37 (m) ^a^
4′–8′	30.3–30.9	1.29–1.36 (m) ^a^	4′–10′	30.4–30.9	1.28–1.39 (m) ^a^	4′–12′	30.3–30.9	1.28–1.37 (m) ^a^
9′	28.1	2.04 (m) ^a^	11′	28.2	2.04 (m) ^a^	13′	28.1	2.04 (m) ^a^
10′	130.8	5.34 (t, 5.1) ^a^	12′	130.7	5.35 (t, 5.1) ^a^	14′	130.8	5.35 (t, 5.1) ^a^
11′	130.8	5.34 (t, 5.1) ^a^	13′	130.7	5.35 (t, 5.1) ^a^	15′	130.8	5.35 (t, 5.1) ^a^
12′	27.9	2.04 (m) ^a^	14′	27.9	2.04 (m) ^a^	16′	27.9	2.04 (m) ^a^
13′	33.1	1.29–1.36 (m) ^a^	15′	33.1	1.28–1.39 (m) ^a^	17′	33.1	1.28–1.37 (m) ^a^
14′	23.4	1.29–1.36 (m) ^a^	16′	23.4	1.28–1.39 (m) ^a^	18′	23.4	1.28–1.37 (m) ^a^
15′	14.4	0.92 (t, 7.1)	17′	14.5	0.92 (t, 7.2)	19′	14.4	0.91 (t, 7.1)

^a^ Overlapping signals; chemical shifts were determined from HSQC and HMBC correlations.

**Table 2 foods-13-01495-t002:** ^13^C (150 MHz) and ^1^H (600 MHz) NMR spectroscopic data of compounds **4**–**7** in CD_3_OD (*δ* in ppm, *J* in Hz).

No.	4		No.	5		No.	6		No.	7	
	*δ_C_*, Type	*δ_H_* (*J* in Hz)		*δ_C_*, Type	*δ_H_* (*J* in Hz)		*δ_C_*, Type	*δ_H_* (*J* in Hz)		*δ_C_*, Type	*δ_H_* (*J* in Hz)
1	201.7		1	199.9		2	161.4		2	161.5	
2	128.0	5.88 (s)	2	128.4	5.91 (s)	3	102.8	6.25 (s)	3	102.8	6.27 (s)
3	167.6		3	164.8		4	106.1	6.84 (d, 2.5)	4	106.1	6.83 (d, 2.5)
4	68.7	4.39 (dd, 8.2, 4.6)	4	74.4	4.17 (dd, 6.8, 1.4)	5	153.9		5	153.9	
5	32.7	1.97 (dtd, 12.8, 8.2, 4.0) 2.23 (ddt, 12.8, 6.8, 4.6)	5	71.0	3.95 (ddd, 9.5, 6.8, 4.3)	6	112.5	6.66 (dd, 8.7, 2.5)	6	112.5	6.65 (dd, 8.7, 2.5)
6	35.5	2.37 (m) 2.54 (dt, 10.8, 6.8)	6	44.4	2.42 (dd, 16.4, 9.5) 2.73 (dd, 16.4, 4.3)	7	111.5	7.15 (d, 8.7)	7	111.5	7.16 (d, 8.7)
1′	43.0	2.44 (dd, 14.2, 4.6) 2.59 (dd, 14.2, 8.0)	1′	42.8	2.46 (dd, 13.8, 4.8) 2.61 (dd, 13.9, 7.8)	3′	131.1		3′	131.1	
2′	71.1	3.86 (hept, 4.5)	2′	72.7	3.87 (hept, 4.5)	7′	150.6		7′	150.6	
3′	38.7	1.50 (m) ^a^	3′	38.5	1.50 (m) ^a^	1″	29.4	2.67 (t, 7.6)	1″	29.4	2.69 (t, 7.6)
4′	26.7	1.50 (m) ^a^	4′	26.7	1.50 (m) ^a^	2″	28.8	1.68 (p, 7.4)	2″	28.8	1.70 (p, 7.4)
5′– 10′	30.3– 30.9	1.28–1.37 (m) ^a^	5′– 12′	30.3– 30.8	1.27–1.35 (m) ^a^	3″– 8″	30.3– 30.8	1.24–1.35 (m) ^a^	3″– 6″	30.3– 30.8	1.26–1.36 (m) ^a^
11′	28.1	2.04 (m) ^a^	13′	28.1	2.03 (m) ^a^	9″	28.1	2.00 (m) ^a^	7″	28.1	2.02 (m) ^a^
12′	130.8	5.34 (t, 5.6) ^a^	14′	130.8	5.34 (t, 5.6) ^a^	10″	130.8	5.32 (t, 5.7) ^a^	8″	130.8	5.33 (t, 5.4) ^a^
13′	130.8	5.34 (t, 5.6) ^a^	15′	130.8	5.34 (t, 5.6) ^a^	11″	130.8	5.32 (t, 5.7) ^a^	9″	130.8	5.33 (t, 5.4) ^a^
14′	27.9	2.04 (m) ^a^	16′	27.9	2.03 (m) ^a^	12″	27.9	2.00 (m) ^a^	10″	27.9	2.02 (m) ^a^
15′	33.1	1.28–1.37 (m) ^a^	17′	33.1	1.27–1.35 (m) ^a^	13″	33.1	1.24–1.35 (m) ^a^	11″	33.1	1.26–1.36 (m) ^a^
16′	23.4	1.28–1.37 (m) ^a^	18′	23.4	1.27–1.35 (m) ^a^	14″	23.4	1.24–1.35 (m) ^a^	12″	23.4	1.26–1.36 (m) ^a^
17′	14.4	0.91 (t, 7.1)	19′	14.4	0.90 (t, 7.1)	15″	14.4	0.89 (t, 7.0)	13″	14.4	0.90 (t, 7.1)

^a^ Overlapping signals; chemical shifts were determined from HSQC and HMBC correlations.

**Table 3 foods-13-01495-t003:** In vitro *α*-glucosidase inhibitory activity (%) of compounds **1**–**18** (50 μM) ^a^.

Compd.	Inhibition Rate ^b^	IC_50_ (μM) ^c^	Compd.	Inhibition Rate ^b^	IC_50_ (μM) ^c^
**1**	−8.1 ± 2.4	-	**11**	43.8 ± 1.3	-
**2**	−12.4 ± 0.66	-	**12**	66.6 ± 0.92	38.3 ± 0.53
**3**	−13.6 ± 1.7	-	**13**	65.1 ± 2.0	36.3 ± 1.3
**4**	11.1 ± 1.1	-	**14**	26.2 ± 1.3	-
**5**	−11.2 ± 0.94	-	**15**	16.6 ± 0.57	-
**6**	97.7 ± 0.19	3.5 ± 0.12	**16**	7.5 ± 1.4	-
**7**	96.7 ± 0.51	2.26 ± 0.06	**17**	37.6 ± 0.12	-
**8**	98.7 ± 0.20	4.5 ± 0.10	**18**	4.8 ± 0.38	-
**9**	61.1 ± 1.6	43.9 ± 0.96	Quercetin ^d^	-	5.2 ± 0.25
**10**	36.4 ± 0.72	-	Acarbose ^d^	-	229.0 ± 0.4

^a^ Data expressed as means ± SD (*n* = 3); ^b^ Inhibition ratio (%) at a concentration of 50 μM; ^c^ IC_50_ = half-maximal inhibitory concentration to *α*-glucosidase; ^d^ Positive control.

## Data Availability

The original contributions presented in the study are included in the article/Appendix A, further inquiries can be directed to the corresponding author.

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
