# Peer review of "Undescribed Cyclohexene and Benzofuran Alkenyl Derivatives from Choerospondias axillaris, a Potential Hypoglycemic Fruit"

_foods, 2024, doi:10.3390/foods13101495_

Round 1

Reviewer 1 Report

Comments and Suggestions for Authors

Overall, the manuscript is well written. In the attached pdf document are the correction suggestions, highlighted in yellow.

It is necessary to take advantage of the ROESY results to assign the stereochemistry on the double bond. Also, you can use the value of the coupling constant of the vinyl protons, so that a value between 16-18 hz is trans, while a value of 10-12 hz is cis. If overlapped, maybe a J-resolved experiment could hepl to solve this problem.

Make an extensive review of the use of significant figures, both in the table and in the text.

Author Response

  1. We gladly accept the comment and appreciate the detailed explanation. Since the two olefinic protons (δH 5.34), as well as the adjacent methylene protons (δH 2.04), are overlapped,  the ROESY correlation couldn’t be applied to resolve the stereochemistry on the double bond. However, as suggested by several references (Queiroz, et al., 2003, Roumy, et al., 2009, Ledoux, et al., 2017, Correia, et al., 2001, Jie, et al., 1997),  when observing the 13C NMR spectrum, the configuration of the double bond in the alkyl chain can be assigned as Z based on the chemical shifts  (δC 27.9 and 28.1) of the adjacent carbon atoms, which would have been more shielded in the case of an E-configuration (δC ∼32). Consecutively,  the isolated components from C. axillaris in the present study, exhibited Z-configuration, based on their upper field chemical shifts of the adjacent carbons to the double bonds. This illustrated from line “236-239”.

Ref: Correia, S. D.; David, J. M.; David, J. P.; Chai, H. B.; Pezzuto, J. M.; Cordell, G. A. Phytochemistry 2001, 56, 781−784.

Jie, M. S. F. L. K.; Pasha, M. K.; Alam, M. S. Lipids 1997, 32, 1041−1044.

Ledoux, A.; St-Gelais, A.; Cieckiewicz, E.; Jansen, O.; Bordignon, A.; Illien, B.; Di Giovanni, N.; Marvilliers, A.; Hoareau, F.; Pendeville, H.; Quetin-Leclercq, J.; Frédérich, M. J Nat Prod. 2017, 80(6), 1750-1757.

Queiroz, E. F.; Kuhl, C.; Terreaux, C.; Mavi, S.; Hostettmann, K. J Nat Prod. 2003, 66(4), 578-580.

Roumy, V.; Fabre, N.; Portet, B.; Bourdy, G.; Acebey, L.; Vigor, C.; Valentin, A.; Moulis, C. Phytochemistry. 2009, 70(2), 305-311.

  1. We have made all the necessary corrections.
  2. Other mistakes were corrected according to the marked PDF file provided by the reviewer.

Reviewer 2 Report

Comments and Suggestions for Authors

Dear authors,

I have reviewed your work submitted to the Foods journal, and I must say that it is an interesting and well-planned study that was carried out with great scientific rigor and robust infrastructure to confirm the presence and biological activity of the molecules. In order to further improve the quality of your manuscript, I recommend the following comments and suggestions:

- The end of the introduction section should present the research objective. However, the authors present a summary of the results instead. I recommend that the authors present the general objective of the manuscript, which should have a direct relationship with the title of the manuscript.

- Can the authors mention the social importance of the fruit? The amount produced? Can the use of fruit as a functional food bring benefits?

- I believe that the information on the maturity of the fruits used in the study is very limited. Do the authors know the maturity index, SST, and acidity? Do they know the colorimetry of the fruits? These data are important because the maturity of the fruits influences the phytochemical profile.

- The authors should justify the use of quercetin as a positive control.

- At the end of the materials and methods section, the authors should present the statistical analyses used, software, calculation of IC50, and repetitions.

- Please review the quality of figure 2.

- The authors should mention in the introduction section if there are any precedents of the use of fruits with hypoglycemic activity in traditional medicine of countries. Can the results of this study confirm their use in traditional medicine?

Best regards.

Author Response

  1. We have made corrections on line “55-59”
  2. We have added the mentioned contents by referring to several literaturesand compiled it on line “36-41”
  3. The studied fruits were collectedon its harvesting stage and started extracting process while its “fresh”, hence we have mentioned this inside the article. Since the manuscript is mainly focused on chemical constituents of the fresh mature fruits, its maturity index, SST and acidity will be measured if further application and production is going to be carried out.
  4. Based on the comment, we have reviewed several articles and incorporated why we have used quercetin as a standard for the α-glucosidase inhibitory activity, line “181-185”
  5. We have added statistical analyses section from line “205-207”
  6. The quality of figure 2 was improved.
  7. As per the comment, we have reviewed several articles and incorporated this section from line “30-33”.